

# Analysing wind and biomass electricity potential in rural Germany considering local demand in 15-minute intervals

Laura Stößel[1], Esther Kohl[1], Björn Roscher[1], Ralf Schelenz[1] and Georg Jacobs[1]

[1]Center for Wind Power Drives, RWTH Aachen, Campus-Boulevard 61, 52074 Aachen, Germany

*Correspondence to*: Laura Stößel (laura.stoessel@cwd.rwth-aachen.de)

**Abstract.** Uncoordinated extension of renewable energy sources (RES) disregarding local demand structures leads to increased loads on the transmission grid and overall economic losses. One approach to solve this problem is to support the local power consumption by local power generation without making use of the transmission network. Therefore, the actual physical coverage of local demand with local supply is to be investigated instead of a yearly net power balance. Rural municipalities

are an ideal starting point to establish such self-sufficient power supply systems on the basis of RES as they have a high RES potential in combination with low demand loads.

Fluctuating feed-in of wind and solar power and peaks in demand loads can be balanced by bioenergy as flexible power generation capacity. In contrast to highly resource dependent wind and solar power, biomass can be stored and power generation from biomass can be controlled flexibly. To assess the potential of electricity from biomass, this study analyses the

agricultural structure of the rural municipalities. The objective of this study is to assess what kind of agricultural structure might be advantageous for flexible power generation from bioenergy, hence balancing fluctuation RES feed-in and power demand. The results from this structural assessment of rural municipalities can help for analysing further municipalities to identify potentials 'at first sight' without costly individual analysis. Heat and fuel sectors are neglected. A methodology is introduced to model time series of wind, PV and biomass power with a 15-minute resolution. It is evaluated to which degree

it is possible to cover the local demand in power supply with bioenergy as flexible power generation capacity in the identified clusters.

The results indicate that bioenergy is generally suitable to cover the gap between local power demand and supply. Waste products from animal farming are far more effective for biomass power production than from agricultural farming. Low population densities raise the potential for self-sufficiency in the power sector because of low demand loads. Further

improvement of the model is needed concerning the clustering approach and for the approximation of installed wind and PV power capacities.

## 1 Introduction

Further expansion of renewable energy sources (RES) is inevitable to reduce emissions in the energy sector and reach the worldwide climate goals. The high renewable energy potential in rural municipalities in Germany plays a crucial role to fully



adopt the concept of RES, or "Energiewende" as known in Germany. In rural areas the renewable energy potential is significantly higher than in more densely populated regions (Jenssen et al., 2012). In Germany many local governments are involved with expanding renewable electricity generation capacities, however these are mostly aiming for energy production and demand balance, disregarding their actual physical coordination (Agentur für erneuerbare Energien, 2019). The gap between electricity generation and consumption is compensated by the electricity network system to obtain security of supply.

With expanding decentralized energy production capacities, this compensation task increases in complexity. Growing shares of decentralized renewable electricity generation capacities that are uncoordinated with the actual local demand loads lead to a stronger, bidirectional interaction with the overarching grid infrastructure (McKenna, 2018). To withstand varying loads, grid modernization measures are inevitable to adapt to the changing electricity system. To relieve the grid and limit costly grid extension, one approach is to support the local electricity consumption of local electricity generation without making use of

the transmission network (Schmidt et al., 2012; Thrän et al., 2015; Umweltbundesamt, 2013). Especially in rural areas, the authors suggest that municipal energy systems strive after a demand-based local electricity supply with regard to the timing of both supply and demand. Relying on the transmission grid should be an exception. The high renewable energy potentials in combination with low electricity demands compared to more densely populated regions make rural municipalities suitable sites to develop self-sufficient electricity systems. Through the establishment of such power supply systems, local potential in

power generation can be exploited efficiently with only limited transmission grid extension. Advantages of a demand-based electricity supply system in rural municipalities include higher acceptance in the population for renewable electricity generation capacitates as well as an acceleration in restructuring the electricity system towards renewable generation capacities (Schreiber, 2012). While this might not lead to the overall economic optimum, advantages in overcoming the constraints of resistance in the population and slow progress in expansion measurements are not to be underestimated (Schreiber, 2012).

Following the distinction between different forms of municipal energy systems and energy autarky by McKenna et al. 2015, this understanding lies between aiming for an annual balance of local generation and consumption and a completely off grid solution (McKenna et al., 2015). The interconnection with the transmission grid persists for times of positive or negative excess power within the municipality.

To increase power self-sufficiency in rural municipalities by coordinating local power generation and demand, there are three

main possibilities: increase of storage capacities, demand-side management and flexible generation capacities. Storage capacities are a promising option, however, quite expensive and still under technical development (Verband deutscher Ingenieure, 2017). Demand-side management is especially useful with large-scale industrial consumers. In rural areas, with the largest consumer group being households, the significance of demand-side management is still object of research. Therefore, in the presented approach the focus is on flexible generation capacities to balance fluctuating RES. With the

advantage of high availability as well as high predictability electricity from biomass can compensate for times when there are little sun or wind resources available. A demand-driven operation of bioenergy generation capacities can significantly reduce daily fluctuations in residual loads (Tafarte et al., 2017; Thrän et al., 2015). Bioenergy has been established in Germany for





years and has the potential to offset fluctuations in solar and wind energy feed-in. In 2018 in Germany the national gross power generation was 542 TWh with biomass contributing 45 TWh (8.3 %) (Fraunhofer ISE, 2019).

Several individual studies conclude that in rural areas power self-sufficiency on the basis of RES is generally possible, however, the economic value has been questioned (Jenssen et al., 2012; Schmidt et al., 2012; Tafarte et al., 2017; Umweltbundesamt, 2013). Especially the further technical development and integration of storage capacities as well as bioenergy as flexible generation capacity is one crucial requirement for self-sufficient power systems (Thrän et al., 2015; Schreiber, 2012; Jenssen et al., 2012; Hahn et al., 2014). Hansen et al. (2019) give a comprehensive overview on most

publications since 2004. The authors conclude that there is a focus on individual country studies with Europe, USA and Australia being well-researched. The need for applying cross-sectoral studies as well as coordinating individual studies with the global context is identified. Also, most studies apply full hourly simulations (Hansen et al., 2019).

The objective of this study is to address some of the identified research needs: This study attempts to determine the status of security of supply in Germany once all rural municipalities have attained their optimal RES potentials, by using a clustering

approach for power potential of biomass as flexible generation capacity and a simulation of wind power, solar and biomass in 15 minute resolution. The high resolution provides self-sufficiency instead of yearly balances. Usually, municipal decision makers are not capable of developing and implementing such power systems due to lack of technical expertise. Technocratic solutions are not suitable in this context either, because a high level of local knowledge is required to ideally exploit local potentials in accordance with the population. To enable municipalities to prepare, make and implement energy system related

decisions more self-sufficiently, the complexity of the energy system needs to be reduced. The presented approach for complexity reduction includes a clustering of rural municipalities in Germany. The aim is to estimate certain potentials easier and faster by matching a municipality to a cluster than conducting a thorough individual potential analysis. In this study, the municipalities are clustered by agricultural structure indicators. Agricultural structure being indicators on the extent of animal and agricultural farming within a municipality. This has a high impact on the availability of biomass which directly influences

the biomass power potential. Given the potential of biomass power as flexible power generation capacity, it is a highly relevant electricity source in the context of municipal power self-sufficiency.

In order to evaluate the potential of power self-sufficiency of rural areas this study attempts to determine the availability of different biomass resources dependent on the agricultural structure. Municipalities with a strong agricultural sector presumably tend to have a higher potential in electricity generation from biomass. The objective is to find patterns and correlations in

agricultural structure and RES potential that are generally valid. Overall the following question is to be answered: What kind of agricultural structure is advantageous for a full cover of local demand by local supply through flexible biomass power production in addition to wind and PV power? The results will enable municipal decision makers to prepare, make and implement energy system related decisions more self-sufficiently.

The agricultural structure assessment is done by a simplified clustering method for rural municipalities in Germany as

described in section 2.1. Time series for both power generation from RES and demand loads are modelled, further described





in section 2.2. In section 3 we present and discuss some results of the analysis. Section 4 presents conclusions and suggestions for next steps.

## 2 Methods and Data

Only the power sector is modelled, heat and fuel sectors are neglected. To pay attention to land use conflicts waste products
only are considered for power form biomass, energy crops are neglected. Rural municipalities up to a population size of 50,000 people are investigated. It is evaluated to which degree it is possible to cover the local demand in power supply with bioenergy as flexible power generation capacity. The bioenergy installations are only operated if the local electricity demand is not covered by local solar and wind power. Consequently, bioenergy is used to minimize the residual load within the municipality. The methodology is divided into two parts. Firstly, a simplified clustering method is applied to provide an analysis of structural
assessment in rural municipalities in Germany (section 2.1). Secondly, time series for electricity generation as well as load profiles for the identified clusters are modelled (section 2.2).

### 2.1 Cluster analysis

The cluster analysis provides a suitable tool to find patterns and correlations in agricultural structure and to assess RES potential which are based on statistical coefficients without the use of time intensive potential analysis tools. This supports the decision-
making for implementing energy systems and to assess the degree of self-sufficiency. On the basis of publicly available agricultural structure data for German districts, a clustering approach was conducted (Statistische Ämter des Bundes und der Länder, 2017; Thünen-Institut für Betriebswirtschaft, 2018b, 2018a; Wirtschaftliche Vereinigung Zucker, 2016).

The clustering is conducted based on five indicators: population density, availability of cattle, pigs, poultry as well as agricultural area for sugar beets. These indicators are selected because of their relevance for power generation from biomass.
The first important indicator in this cluster analysis is the population density as it is presumed to be functionally related with the highest RES potential. A measure of population density is provided by Bundesinstitut für Bau-. Stadt- und Raumforschung (2011). In Germany, 90 % of the total area is considered rural that includes $46.9*10^6$ people (Bundesministerium für Ernährung und Landwirtschaft, 2015). However, the smooth transition from urban to rural regions makes a clear definition difficult. Generally, a population density higher than 845 citizens per square kilometre counts as non-rural area (Bundesministerium für
Ernährung und Landwirtschaft). All clusters are to be categorized as rural and are below the German average of 237 citizens per square kilometre (Statistisches Bundesamt, 2018a).

A minimum level of biogenic waste product availability is necessary to consider power production from biogenic waste products a worthwhile option. Therefore, only areas with at least average agricultural and/or animal farming are taken into account and the average is determined in accordance with the data basis (Statistische Ämter des Bundes und der Länder, 2017;
Thünen-Institut für Betriebswirtschaft, 2018b, 2018a; Wirtschaftliche Vereinigung Zucker, 2016).



The identified rural municipalities in Germany are filtered according to their agricultural and animal farming characteristics. All districts with above average cattle availability are identified (Statistische Ämter des Bundes und der Länder, 2017). For those regions the availability of the other categories is determined. This process is iteratively repeated for pig, poultry and sugar beet availability.

**2.2 Simulation methodology**

To assess the possibilities for flexible power generation from bioenergy, the power sector needs to be modelled as a first step. Since it is aimed for 100 % RES in the power sector of rural municipalities, relevant generation capacities include wind, PV and biomass power. In rural contexts, hydroelectric power has been neglected as these power plants usually are connected to the transmission grid due to high rated power and therefore only play a minor role in decentralized power systems (Verband
deutscher Ingenieure, 2017).

The wind power feed-in is modelled by the analytic tool WIFO (Wind Farm Optimization). The tool was developed at Center for Wind Power Drives and is capable of determining the power feed-in for wind turbine generators (WTG) in any arbitrary wind farm layout (Roscher et al., 2018a; Roscher et al., 2018b). For modelling, the reference site according to German Renewable Energy Sources Act is chosen (Bundesministerium der Justiz und für Verbraucherschutz, 2017). WiFO is run for
six 4-MW-WTG at the site with a given time series of wind speeds in 2017 provided by the German Weather Service (Deutscher Wetterdienst, 2019). The model results are then scaled to the assumed installed capacity in the respective investigated municipality.

Solar power time series are approximated on the basis of Fraunhofer Energy Charts (Fraunhofer ISE, 2019). The installed capacity for the analysed municipality is scaled down from the overall installed capacity in Germany according to the municipal
area.

Power feed-in from biomass is simulated in a newly developed potential analysis tool. On the basis of individual characteristics of a municipality the potential for power generation from biogenic waste products is assessed. Relevant waste products from animal farming include liquid and solid manure from cattle, pigs and poultry. Other animal groups can be neglected as they represent less than 5 % of the total livestock in Germany (Landesamt für Natur and Umwelt und Verbraucherschutz NRW,
2014). Regarding relevant agricultural waste products in Germany leaves from sugar beet are considered. In theory, straw and potato plants could contribute relevant mass influxes. However, technologies for power generation from straw are still under development and potato leaves are difficult to collect (Kaltschmitt et al., 2016). Accordingly, straw and potato plants can be neglected in this context. In addition, wooden biomass and green waste from forest or grounds maintenance as well as scrap from industry and private sector are considered. Finally, municipal waste products are taken into account.

Based on the input parameters population size, area size as well as livestock within the municipality, the biomass availability for each month is calculated in the model. Waste products from animal farming are calculated on the basis of available livestock under consideration of standard allocation factors (Deutscher Bundestag, 2017; Kuratorium für Technik und Bauwesen in der Landwirtschaft, 2012; Statistisches Bundesamt, 2018b, 2019). Wooden biomass and green waste from forest or grounds





maintenance are approximated on the basis of common land use shares in Germany (Statistisches Bundesamt, 2018c). For
scrap from industry and private sector as well as municipal waste products mean factors for distribution based on population
and area size respectively are used (Kaltschmitt et al., 2016). The collected waste products are then combusted for electricity
generation, either after transformation into biogas or as solid fuel.

In a second step, it is assessed whether the biomass mass flow exceeds the capacities of the existing bioenergy generation
capacities in the municipalities. If so, suggestions for new generation capacities are developed. Depending on the kind of
excess biomass (solid or biogas substrate), CHP (combined heat and power plant) with either gas or steam turbine are chosen.
This is determined under consideration of a flexible operation of all bioenergy capacities. Consequently, electricity from
biomass is used to balance offsets between wind and PV power feed-in and demand.

As a result, time series for potential power production from biogenic waste products are generated. In combination with feed-
in time series for wind and PV power from the other modules, the potential renewable power supply for a municipality is
simulated. Finally, the load profile for the municipality is approximated on the basis of standard load profiles provided by the
Federal Association of Energy and Water Economics in Germany (Bundesverband der Energie- und Wasserwirtschaft, 2018).
Considering the local demand load profile in a 15-minute time resolution, the power production for a year is modelled.

To assess the degree of self-sufficiency, meaning the coverage share of local electricity demand with local generation units,
the feed-in and demand time series are compared. It is evaluated in how many time steps, the local feed-in covers or exceeds
the local demand load (compare section 3).

## 2.3 Modelling and scaling time-series

To combine the clustering approach with the simulation methodology, the three RES feed-in time series – wind, PV and
biomass power – as well as the demand load is calculated. Wind and PV power are quite roughly approximated (compare
section 2.2). Wind power is simulated for one wind farm with 24 MW installed capacity on the reference site according to
German Renewable Energy Sources Act (Bundesministerium der Justiz und für Verbraucherschutz, 2017). The PV power
feed-in time series is approximated on the basis of Fraunhofer Energy Charts for the whole of Germany (Fraunhofer ISE,
2019). Both time series are calculated for a certain installed capacity. Depending on this basis, the feed-in series are scaled to
the installed capacity in the municipal clusters, depending on the respective cluster area.

Wind and PV power in the respective municipalities are scaled depending on the available area size following Eq. (1):

$$C_{mun} = A_{\mathrm{mun}} * C_{Ger} \, , \tag{1}$$

Where $C_{mun}$ represents the installed capacity in wind and PV power in the respective municipality and $C_{Ger}$ represents the
overall installed capacity in wind and PV power in Germany. $A_{\mathrm{mun}}$ is the municipal area share on the whole of Germany.

In Germany about 46 GW installed PV power and 53 GW installed onshore wind power are taken as a basis (Fraunhofer ISE,
2019). The total area of Germany amounts to 357 582 square kilometres. With this approach, regional differences in resource
availability for wind and PV power are neglected.



The biomass power time-series is modelled more individually depending on the biomass availability in the respective clusters. As explained above, waste products only are considered, energy crops are neglected (compare section 2.2).

The demand load profile is approximated on the basis of standard load profiles provided by the Federal Association of Energy and Water Economics in Germany (Bundesverband der Energie- und Wasserwirtschaft, 2018).

## 3 Results and Discussion


In the following, exemplary results are presented and discussed. It is important to note, that this is not applied to a yearly net balance but to a 15-minute-resoluted time series. Only with this method, the actual degree of self-sufficiency can be determined. A yearly net balance rather increases the bidirectional interaction with the overarching grid infrastructure (McKenna, 2018). Thus, a 50 % share of coverage means, that half of the time over the modelled year, the local power demand

is covered with local power supply. Times of excess power feed-in are set to a 100%-coverage and it is assumed that excess electricity is fed into and gaps in supply are covered by the transmission grid.

### 3.1 Cluster analysis

The cluster analysis results in five different clusters. The characteristics of the clusters are listed in Table 1 and the respective location can be seen in Figure 1: Spatial distribution of identified clusters.Figure 1. The clusters are located in regional groups.

However, this will probably change with further improvement of the method, especially by integrating further criteria for clustering.

**Table 1: Characteristics of identified clusters.**

| Cluster | Population density [citizens/km²] | Cattle/ citizen [-] | Pigs/citizen [-] | Poultry/ citizen [-] | sugar beet area / agricultural area [km²/km²] |
|---|---|---|---|---|---|
| A | 113.05 | 0.72 | 5.09 | 70.76 | 0.01 |
| B | 130.21 | 0.36 | 2.38 | 1.06 | 0.01 |
| C | 157.21 | 0.72 | 0.51 | 1.13 | 0.02 |
| D | 65.28 | 0.34 | 1.00 | 7.51 | 0.05 |
| E | 153.09 | 0.22 | 0.03 | 0.77 | 0.20 |





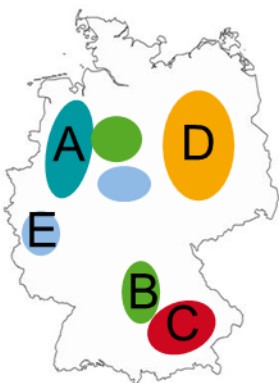


**Figure 1: Spatial distribution of identified clusters.**

Cluster A shows the highest animal farming density with population density being average and sugar beet cultivation being low. Average values for all indicators are calculated on the basis of all rural municipalities. Cluster B is strong only in pig farming, all other categories are average or below. In contrast, cluster C is strong only in cattle farming, all other categories are average or below. Cluster D stands out in that effect that all biomass resources show an average availability with the

population density being below average. Finally, cluster E is the only cluster with significant sugar beet cultivation. Simultaneously the population density is comparatively high; all animal farming categories are below average.

As a result of this clustering, five categories with all relevant information to derive the biogenic waste products mass flow for power production as well as time series for PV and wind power production are determined (compare sections 2.2 & 2.3).

The determined installed capacities for RES are listed in Table 2.


**Table 2: RES installed capacities for identified clusters.**

| Cluster | Municipal area [km²] | Installed capacity PV [MW] | Installed capacity wind [MW] | Installed capacity biomass [MW] |
|---|---|---|---|---|
| **A** | 2882.00 | 370.18 | 427.24 | 60 |
| **B** | 1484.00 | 190.61 | 220.00 | 21.12 |
| **C** | 870.72 | 111.84 | 129.08 | 16.47 |
| **D** | 1930.00 | 247.90 | 286.11 | 18.87 |
| **E** | 1248.73 | 160.39 | 185.12 | 17.76 |

### 3.2 Seasonal analysis

With wind and PV power being the main contributors to power supply, not only daily but also seasonal variations need to be taken into account. In times of high wind and PV power feed-in, bioenergy is less relevant than in calm nights, for example.

Therefore, these variations need to be analysed as well.

Taking a look at Figure 2, exemplary time series of one week in January and July in cluster D, respectively, can be seen. In the exemplary January week (Figure 2 (a)), Wednesday early morning through Thursday night, feed-in rates from solar and



wind are quite high, clearly dominated by wind power. In the exemplary July week (Figure 2 (b)), Friday through Sunday little to no wind power is fed-in. The degree of self-sufficiency over the whole week is similar for January (46.1 %) and July (46.9

%). However, the distribution is quite different. The seasonal differences underline that a sole extension of solar and/or wind power cannot provide security of supply. While in summer, when PV power is strong, demand and supply curves correlate clearly, in winter there is little to no correlation. Thus, flexible power generation from biomass is of high importance to balance offsets between power supply and demand. However, the installed capacities in the biomass model are not significant enough to make a difference. Furthermore, wind and solar power alone rarely get close to covering the demand load. Accordingly, an

overall extension of electricity generation capacities in the model is needed. Since the installed capacities are approximated quite roughly, a sophistication of the model is definitely required here. While wind and PV power capacities are estimated, biomass installed capacities are calculated on the basis of biomass availability from biogenic waste products. These installed capacities are much lower than wind and PV power (compare Table 2). Although, this is a realistic assumption, biomass' capability to balance fluctuating wind and PV power feed-in tends to be highly limited with energy crops being neglected.

Accordingly, the integration of further biomass sources – within a reasonable scope - is a next step to improve the model. Furthermore, storage capacities and ways of demand-side management should be considered to smoothen peaks in demand.



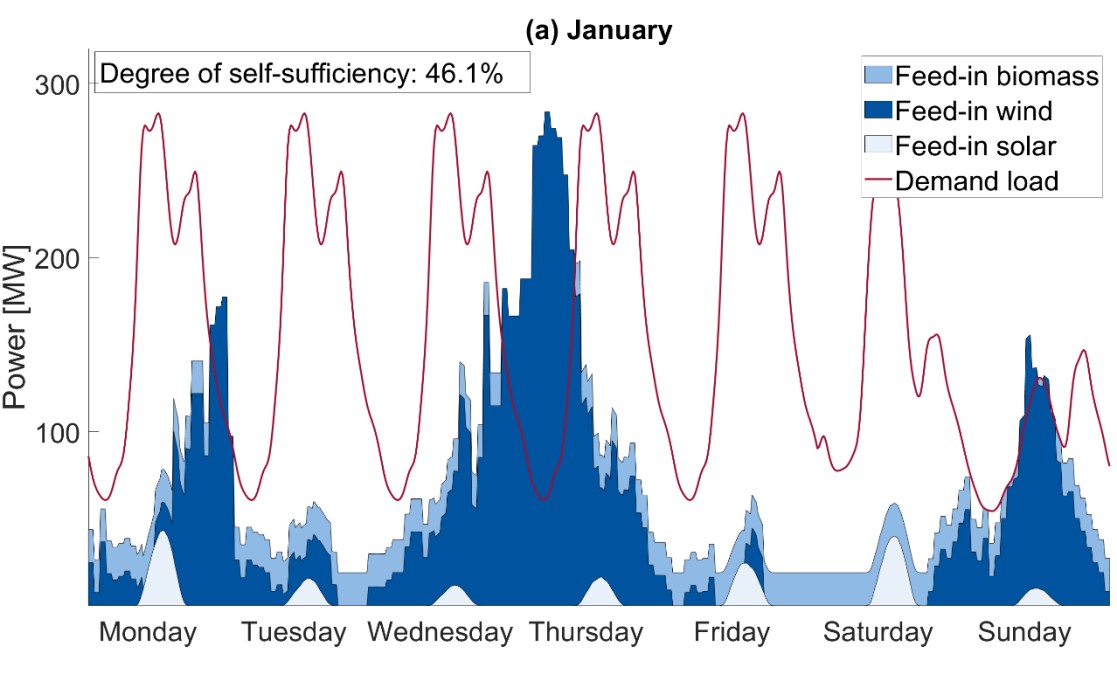

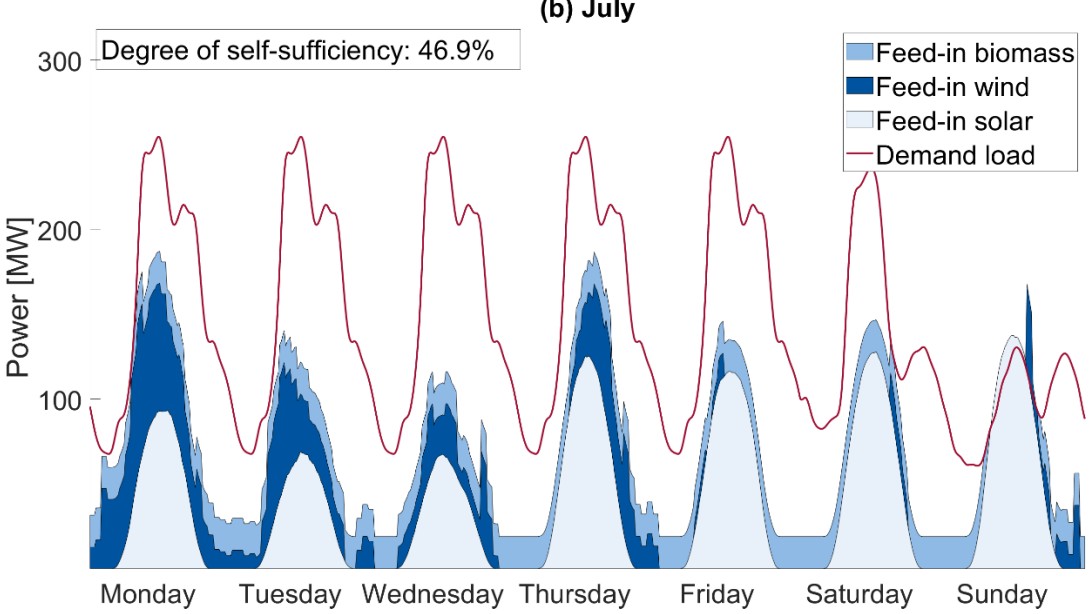

**Figure 2: Exemplary time series of one week in January (a) and July (b) for cluster D.**


In Table 3 the level of self-sufficiency in the power sector for exemplary months in winter (January) and summer (July) is compared. Obviously, this only covers the example of one month of one year and especially in times of changing seasonal



conditions, this only shows a section. For a more representative statement, further years need to be modelled. However, it gives an indication of the underlying pattern.

In July a slightly higher degree of self-sufficiency is reached than in January. This applies to all clusters but cluster E. Furthermore, cluster C only shows a small difference in self-sufficiency in January and July. This might be explained by both the comparatively small municipal area and the high population density of these two clusters. As installed capacities in wind and solar power are scaled according to the available area, the installed capacities in comparison to the demand loads remain small (compare section 2.3). Following, the seasonal difference in wind and solar power is not as pronounced in relation to the

demand loads.

The tendency for higher self-sufficiency in July than in January can be explained by a certain constancy PV power provides. While PV power by its very nature is limited to daytime, there is always some feed-in, even on cloudy days. This diurnal dependency does not apply to wind, so that times of high wind speeds can result in high wind power feed-in 24/7. On the other hand, times of low wind speeds can also result in little to no wind power feed-in. In this regard wind power is less reliable than

PV power.

**Table 3: Level of self-sufficiency in the power sector comparing winter and summer for each cluster**

| Cluster | January [%] | July [%] |
|---------|-------------|----------|
| A | 45.80 | 47.58 |
| B | 36.71 | 37.37 |
| C | 36.23 | 36.28 |
| D | 43.47 | 47.18 |
| E | 34.62 | 34.14 |

### 3.3 Year-round analysis

In Figure 3, box plots for each cluster are displayed. The red middle line represents the median value, the box the upper and lower quantiles and the surrounding bars show the overall value distribution. It can be seen that all municipalities scatter quite

strongly. This means that self-sufficiency cannot be provided at a constant level. Besides, the overall degree of self-sufficiency is quite low in all clusters.



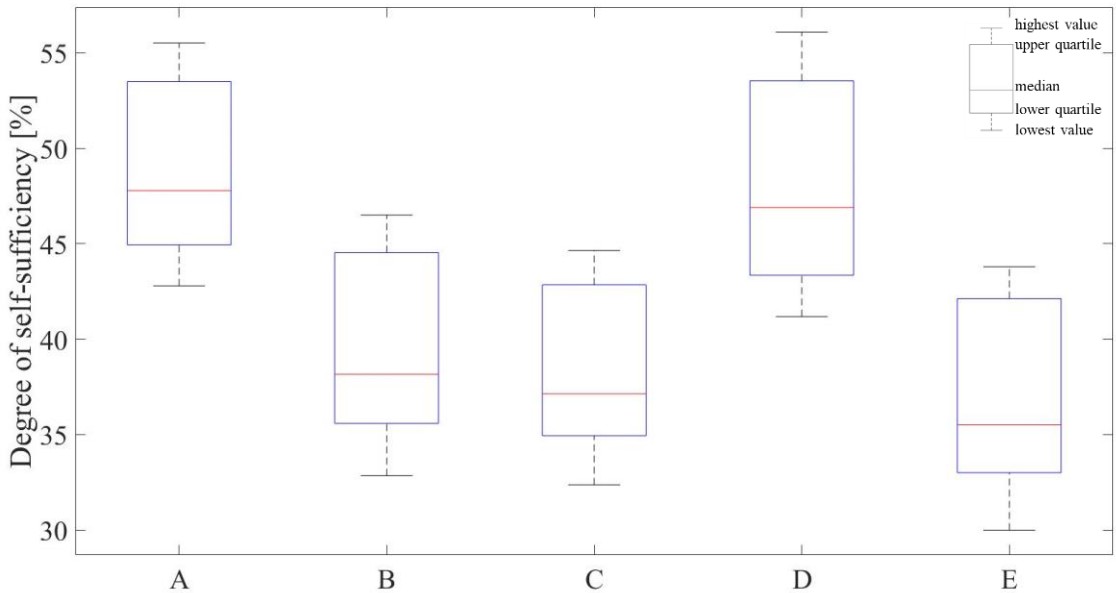

**Figure 3: Boxplots for distribution of level of self-sufficiency over the course of the year for each cluster**

Table 4 shows the level of self-sufficiency in the power sector for each cluster in the whole year. None of the five clusters can

reach electricity autarky even half of the time. The lowest coverage is found in Cluster E with 37 %. Cluster E is the only
cluster with significant sugar beet cultivation. In Germany, intensive sugar beet cultivation is concentrated in specific areas.
These areas usually do not provide extensive farming of other cultures or animals (Wirtschaftliche Vereinigung Zucker, 2016).
Accordingly, the only relevant biogenic waste source are leaves from sugar beet cultivation. Also these regions are usually
characterised by a relatively high population density, which results in higher and more complex demand load structures. These

cluster characteristics explain the low share of local demand by local supply in cluster E.

Clusters B and C can only reach marginally higher coverage shares. The municipalities within these clusters all are moderately
densely populated (130-157 citizens/km², compare Table 1) and have low shares of sugar beet cultivation and poultry farming.
Cluster B is strong in pig farming and cluster C in cattle farming. Even though, in the model, cattle provide the most biogenic
waste per animal with an average possible electricity generation of 2613 kWh/animal/year, cluster B shows a slightly higher

coverage share. On average, biogenic waste from pig farming only reaches 504 kWh/animal/year. However, pigs are usually
cultivated in much larger scales. As a result, regions with relatively high pig farming make use of scale effects. For example,
an average reference value for pig farming would be 1 pig per citizen whereas for cattle the average value would be only 0.36
cattle per citizen. This leads to a higher availability of biogenic waste for power generation in municipalities that are strong in
pig farming compared to those strong in cattle farming. Consequently, the potential for offset balancing between feed-in from

wind and PV power and demand loads is higher in cluster B.



The highest share is attained in Cluster A with 49 %, followed closely by cluster D with 48 %. The large advance in coverage for these two compared to the other clusters needs to be explained separately. The only similarity is the high area size in both clusters. As the installed capacity in wind and PV power is scaled according to the available area size, this leads to high installed capacitates to start with. In combination with average to low population densities the initial situation for a self-

sufficient power sector is advantageous. Cluster A includes municipalities that are very strong in all relevant groups of animal farming. Thus, in these municipalities a high amount of biogenic waste is accumulated. This makes it easier to balance offsets between PV and wind power feed-in and local demand. As for cluster D, the municipalities accumulate average biogenic waste in all relevant categories and have the lowest population density among all clusters. The demand loads dependent on the population size, result in municipalities with low demand loads. Consequently, less provided electricity is required to cover

the loads and wind and PV power don't need as much support from bioenergy.

**Table 4: Level of self-sufficiency in the power sector over the course of the year**

| Cluster | [%] |
|---------|--------|
| A | 49,02 |
| B | 39,68 |
| C | 38,60 |
| D | 48,12 |
| E | 37,01 |

## 4 Conclusion and Outlook

In this paper clusters of rural municipalities are investigated to design individual power supply systems on the basis of RES. A methodology is introduced to model time series of wind, PV and biomass power in 15-minute resolution. Also a clustering

approach is presented to categorize rural municipalities in Germany. The goal of the study is to assess what kind of agricultural structure is advantageous for flexible power generation from bioenergy, hence balancing fluctuating RES power feed-in and demand. The clustering approach is focused on agricultural structure data because this is where the biomass for power generation is coming from. The authors want to find patterns and correlations in agricultural structure and RES potential that are generally valid. The results from this structural assessment of rural municipalities can help for analysing further

municipalities to identify potentials 'at first sight' without costly individual analysis. The approach presented is to be regarded as initial clustering that is to be sophisticated in further studies.

The results indicate that bioenergy is generally suitable to cover the gap between local power demand and supply. There are two main takeaways to answer the initially posed question of advantageous agricultural structure for power self-sufficiency. Firstly, waste products from animal farming are far more effective for biomass power production than from agricultural

farming. Secondly, low population densities raise the potential for self-sufficiency in the power sector because they result in



low demand loads that are comparatively easy to cover. The initial hypothesis, that municipalities with a strong agricultural sector presumably tend to have a higher potential in electricity generation from biomass, is corroborated.

However, the approximated installed capacities in the model for all RES are underestimated. Especially wind and PV power are approximated and for bioenergy an integration of energy crops has to be considered to increase the potential. These parts

of the model need to be further investigated. Also the clustering approach is rather qualitatively and the clustering parameters are limited. The integration of further parameters as well as the application of an established clustering methodology is to be applied in future works. Among others, additional indicators include installed capacities of wind and PV power in rural municipalities in Germany as well as socio-technical data. Finally, a model part to integrate storage capacities is to be added to enable smoothening of peaks in demand loads.

After model sophistication, a further application of the methodology is needed for testing and validation. Long-term goal is to cluster all municipalities in Germany and to apply a structure analysis. With these results, an overall prediction on strengths and weaknesses of the German power sector in case of completed optimization in rural areas might be possible.

**Achknowledgements**

We thank the Federal Ministry for Economic Affairs and Energy for funding the research project (acronym: ArkESE) in which

we address this topic (enargus, 2018). In the three-year project (start December 2018), rural municipalities are investigated to design individual electricity supply systems on the basis of renewables. Municipalities shall be enabled to prepare, make and implement energy system related decisions more self-sufficiently.

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
