# Peer review of "Analysing wind and biomass electricity potential in rural Germany considering local demand in 15-minute intervals"

_Wind Energy Science, 2019_

## Referee Comment (RC1) · Anonymous Referee #1 · 24 Jan 2020

This paper focuses on using biomass to balance peaks in wind and solar power, by analysing the agricultural structure of rural municipalities. Whilst this topic is interesting for the wind energy community, and the quality of the work is good, the research work is strongly focused on biomass, and not on wind energy. Therefore the paper is not suitable for this journal and I would suggest submitting it to a general renewable energy journal.

---

## Referee Comment (RC2) · Anonymous Referee #2 · 9 Feb 2020

Analysing wind and biomass electricity potential in rural Germany considering local demand in 15-minute intervals Summary The paper provides a method to analyze the availability of "renewable energy resources" to balance the fluctuations of wind power in demand loads. The method is based on a cluster analysis of the farming-characteristics of rural municipalities in Germany and uses indicator variables such as population density, availability of cattle, pigs, poultry and agricultural area for sugar beets. The cluster analysis is restricted to those indicators which are associated with power generation from biomass, which encompass liquid and solid manure from cattle, pigs and poultry as well as leaves from sugar beet. The comparison of the time-series of the power from the biogenic waste, the time series for wind and PV power enables

the simulation of the potentially renewable power supply for the municipalities. The wind power feed-in is modeled by the analytic tool WIFO (Wind Farm Optimization), which enables to determine the power feed in for wind turbine generators. The model is run for 140 six 4-MW-WTG at the sites and a given time series of wind speeds of 2017, which were provided by the German Weather Service. These are scaled to the assumed installed capacity in the investigated municipality. Solar power time series base on Fraunhofer Energy Charts and were scaled down from the overall installed capacity in Germany according to the municipal area.

The results of the analysis of the different time series of power from biogenic waste, wind power and PV indicate that different degrees of self-sufficiency for different rural municipalities and for different months (January and July). Based on the analysis of the "energy mix" inferences on the suitability of municipalities for self-sufficiency and for future potentials are drawn.

General comments

The paper is well written and the method may improve our knowledge concerning the concepts of self-sufficiency and may provide an important method for decisions associated with a change of the power generation by renewable energy resources. The advantage of the method is that it provides different settings for renewable energies on a regional basis and that it provides a method for the analysis of regional differences of the potentials and therefore supports the choice suitable sites. Although the combined use of wind power, solar energy and bioenergy is not a new idea, the combination of the methods is new. These include the 15 minute resolution time series of wind power, solar power and bioenergy, and the methods for detecting potential areas for self-sufficiency. The results of this approach are a partly preliminary and provisional but appear to be able to indicate regional and temporal differences self-sufficiency for different rural municipalities. Methods and results are clearly presented and supported by figures and tables. The number of references and quality are appropriate. The paper represents a substantial contribution as it is based on a new method which enables the analysis of regional differences concerning the issues related to the potential offsets fluctuation in wind energy feed-in.

However, some parts of the paper may be/ or need to be improved. I have made some specific comments which are aimed to improve the paper and the impact of the inferences and conclusions. - Chapter 3.1 - Cluster analysis - In order to interpret the value of the data more information on variability within the clusters (A, B, C, D) are necessary. The cluster sites encompass very large areas and are characterized by differences in the infrastructure and different types of land use. Somewhat more information concerning the variability of the roughly classified regions appears necessary. In addition some trends or tendencies in the temporal availability of wastes such as liquid and solid manure from cattle, pigs and poultry may indicate whether the amount are increasing or declining within the last decade. Such information may be also important for the calculation of the potential use in the near future.

- In this study the self-sufficiency depends largely on the amounts of biogenec wastes. However, the amounts are subject to economic and ecologic decisions. Liquid manure is often exported to regions with poor soils, where it is used as fertilizer. Thus there is a completion between its use for bioenergy and for its use as fertilizers. Future trends and the turn to a more sustainable ecological agriculture may reduce large-scale animal husbandry and hence reduce the amounts and presumably the availability of wastes such as liquid and solid manure. What are the perspectives in this case? These issues may discussed in a separate (discussion ?) chapter. The paper can be improved by a separate discussion chapter, where these aspects can be discussed more comprehensively.

Additional aspects: - Although the paper is pointing on the role of renewable energy, the analysis is within the scope of wind energy as the major issues are associated withe the balance the fluctuations of wind power in demand loads and the simulations of wind power in time series is concerned to indicate temporal and regional imbalances. - The concepts and ideas are novel and the objectives of the hypotheses are clearly

formulated and put forward. - Results are sufficient and support the interpretation (see my comments to chapter 3.1) - The structure of the paper may be improved by separating results and discussions (see my comments above)

Decision : good (2) with some moderate minor changes (see my comments).

---

## Editor Comment (EC1) · Nicolaos Cutululis (Editor) · 2 Mar 2020

Dear authors,

As mentioned in the reviewers, the paper is well written and has scientific merit. However, with the main subject of the paper being an analysis electricity potential from biomass, the paper is probably best suited in a different journal, focusing on renewable energy in general.
* * *

---

## Author Comment (AC1) · 10 Mar 2020

Dear referees,

thank you very much for your feedback.

You are right, the focus of the paper lies on power production from biomass not wind power. However, as the main problem of fluctuating feed-in from wind power is addressed, we still consider it a good fit for this journal. Furthermore, this is the first publication of the research project and modeling feed-in from wind power will be more strongly emphasized in future works.

A sophistication of the cluster analysis including the integration of further parameters is planned for future works. This will automatically refine the resulting clusters. As for temporal availability of biomass, at the current state of the model only seasonal effects over the course of the year are considered. For future works both the past and the future development of biomass availability will be integrated. This will provide information on future potential development. However, as this requires a thorough research on current trends in agricultural farming in different regions, we consider it out of scope for this paper.